# Biochemical Biomarkers of Mucosal Healing for Inflammatory Bowel Disease in Adults

**DOI:** 10.3390/diagnostics10060367

**Published:** 2020-06-02

**Authors:** Małgorzata Krzystek-Korpacka, Radosław Kempiński, Mariusz Bromke, Katarzyna Neubauer

**Affiliations:** 1Department of Medical Biochemistry, Wroclaw Medical University, Chalubinskiego 10, 50-368 Wroclaw, Poland; mariusz.bromke@umed.wroc.pl; 2Department of Gastroenterology and Hepatology, Wroclaw Medical University, Borowska 213, 50-556 Wroclaw, Poland; radoslaw.kempinski@umed.wroc.pl

**Keywords:** mucosal healing, inflammatory bowel disease, ulcerative colitis, Crohn’s disease, fecal calprotectin, biomarker, endoscopy, c-reactive protein

## Abstract

Mucosal healing (MH) is the key therapeutic target of inflammatory bowel disease (IBD). The evaluation of MH remains challenging, with endoscopy being the golden standard. We performed a comprehensive overview of the performance of fecal-, serum-, and urine-based biochemical markers in colonic IBD to find out whether we are ready to replace endoscopy with a non-invasive but equally accurate instrument. A Pubmed, Web of Knowledge, and Scopus search of original articles as potential MH markers in adults, published between January 2009 and March 2020, was conducted. Finally, 84 eligible studies were identified. The most frequently studied fecal marker was calprotectin (44 studies), with areas under the curves (AUCs) ranging from 0.70 to 0.99 in ulcerative colitis (UC) and from 0.70 to 0.94 in Crohn`s disease (CD), followed by lactoferrin (4 studies), matrix metalloproteinase-9 (3 studies), and lipocalin-2 (3 studies). The most frequently studied serum marker was C-reactive protein (30 studies), with AUCs ranging from 0.60 to 0.96 in UC and from 0.64 to 0.93 in CD. Fecal calprotectin is an accurate MH marker in IBD in adults; however, it cannot replace endoscopy and the application of calprotectin is hampered by the lack of standardization concerning the cut-off value. Other markers are either not sufficiently accurate or have not been studied extensively enough.

## 1. Introduction

Inflammatory bowel disease (IBD), encompassing ulcerative colitis (UC) and Crohn’s disease (CD), are lifelong, potentially devastating conditions of the gastrointestinal tract, and are characterized by a relapsing–remitting course and heterogeneous clinical presentation with extraintestinal manifestations. Although their pathogenesis remains not fully elucidated, a crucial role seems to be played by an interaction of genetic, epigenetic, immunological, and environmental factors. The uncertainty and complexity of IBD pathogenesis result in the absence of a single diagnostic tool, and what is the most vital, the absence of an effective causative therapy [1,2]. This, in turn, drives the search for better therapeutic goals, and in parallel, the dynamic development of new therapeutic agents. In this respect, mucosal healing (MH) in IBD is gaining increasing attention. MH seems to be a better prognostic indicator of the disease outcome compared to clinical scores. In UC, it is associated with a lower risk of clinical relapse, hospitalization, need for immunosuppression and colectomy, and colitis-associated neoplasia [3,4,5,6,7,8]. In CD, MH has been significantly associated with less severe inflammation after five years, a decreased risk of future steroid treatment, and lower rates of surgical resection [4,9,10,11]. The importance of achieving MH during therapy has been repetitively shown in numerous clinical trials, such as ACCENT [12], SONIC [13], and EXTEND [14].

IBD activity can be evaluated on several levels: clinical, biochemical, macroscopic (endoscopy), and microscopic (histology). However, clinical remission in an IBD patient does not necessarily imply MH, and vice versa, some patients with no visible inflammatory changes on an endoscopy report gastrointestinal symptoms, probably due to concomitant functional disorder of the gut. Histological remission is difficult to assess, especially in CD, in which the inflammation may be patchy and the lesions are not continuous (skip lesions) and are thus easily missed by the biopsy. Moreover, in the case of no visible macroscopic changes, it is hard to establish a representative tissue to be sampled for biopsy to confirm histological healing of the mucosa. The natural way to assess the effectiveness of the therapy is by confirming the resolution of lesions via endoscopy. Indeed, the simplest and most common MH definition applied in clinical practice is the absence of inflammation as judged by endoscopists [15]. Many scales have been devised for the objective classification of endoscopic findings. The most often used score for the evaluation of treatment efficacy in clinical trials, although not validated, is the Mayo Endoscopic Score (MES) [16,17], where MH is defined as MES ≤ 1. However, the guidelines of the European Crohn’s and Colitis Organization, as well as the Japanese Society of Gastroenterology, restrict complete endoscopic remission to a score of 0 (normal or completely healed mucosa) [18,19,20]. The only two indices that received formal validation in UC are the Ulcerative Colitis Endoscopic Index of Severity (UCEIS) [21] and the Ulcerative Colitis Colonoscopic Index of Severity (UCCIS) [22]. It ought to be stressed that the application of different indices and the lack of standardization concerning the MH definition complicates the interpretation of data reported by various authors. A summary of the most important endoscopic scales for the evaluation of disease severity in UC is shown in Table 1.

In contrast to UC, inflammatory changes in CD can also appear proximal to the terminal ileum. Moreover, a deeper layer of the gut wall can be involved with the formation of fistulas. The evaluation of the upper digestive tract, small intestine, and non-luminal CD can be performed with upper endoscopy, imaging techniques (magnetic resonance/computed tomography), or with capsule endoscopy. It must be highlighted, that non-invasive and widely available ultrasonography, namely simple intestinal ultrasonography or ultrasonography with contrast enhancement, is a valuable diagnostic tool for evaluating the disease activity and extent [18]. A standard ileocolonoscopy allows for the evaluation of injuries in the luminal mucosa and the detection of potential stenosis. Endoscopic activity may be reliably scored using either the Crohn's Disease Endoscopic Index of Severity (CDEIS) [23] or the Simple Endoscopic Score for Crohn's Disease (SES-CD) [24]. Both scores have been prospectively validated and are highly reproducible with excellent inter-observer agreement. A CDEIS score of < 3 is defined as mucosal remission [25]. The calculation of the score is complicated and time-consuming, which impedes its application in daily practice. SES-CD is a simplified version of CDEIS, making it easier to calculate. It has been validated and compared to CDEIS, with which it has shown a strong positive correlation (*r* = 0.938) [25]. Additionally, the SES-CD is correlated with both clinical parameters and inflammatory indices (C-reactive protein (CRP)) [26]. For clinical trials, an SES-CD value can be converted into a CDEIS value using the following formula: CDEIS = 0.76 × SES-CD + 0.29 [24].

While remaining a golden standard in MH detection, endoscopy is an invasive technique that risks bowel perforation and is poorly accepted by patients. Moreover, it is not easily available, requiring both expensive equipment and an experienced endoscopist. As such, surrogate markers allowing for a non-invasive and inexpensive, but equally accurate, evaluation of mucosa are intensively searched for. Our goal was to provide an overview of MH markers in colonic IBD that have emerged during the last decade, as well as a summary of their performance to address the question of whether they are ready to replace endoscopy in MH evaluation. Collected evidence shows that fecal calprotectin remains the closest to a “perfect” MH marker; however, the application of which is hampered by the lack of standardization concerning the optimal cut-off value. Although some of the emerging MH markers seem to be promising, confirmatory studies and validation are needed.

## 2. Methods

A systematic review was conducted in March 2020 and included manuscripts published from January 2009 in the English language. Through their respective websites, Pubmed, Web of Knowledge, and Scopus databases were queried with a set of 11 inclusion expressions. To prepare queries, the following expressions “mucosal healing”, “inflammatory bowel disease”, “inflammatory bowel diseases”, “ulcerative colitis”, “Crohn’s disease”, “Crohn”, “mucosal inflammation”, “bowel inflammation”, “colonic inflammation”, and “colonoscopy” were used and each of them was matched through an operator AND with a “marker” expression. Query results were cross-searched and cleaned of duplicates. In the following title screening step, publications concerning non-IBD, pediatric IBD, non-colonic CD, unspecified IBD, experimental studies (animal and in vitro), microbiota-related, hematological, on tissue-based markers, on non-MH markers, and non-original articles were excluded. The same criteria were applied for the abstract screening, and finally, for the full-text screening. At the full-text screening step, an additional description of the markers’ performance, in terms of their accuracy and/or sensitivity and specificity, was required.

At each step, the article selection was verified by a second investigator. No authors were contacted for further data. The following data were retrieved from the reviewed publications: the type of potential marker and its source (serum/plasma, feces, urine), the method of assessment and assay manufacturer (in the case of calprotectin), the IBD phenotype (CD or UC, or mixed IBD cohort), study population including the number of patients with MH, additional characteristics of the evaluated cohort if specific (clinical remission, treatment), the score used for the evaluation of endoscopic findings and the MH definition applied, the correlation between the evaluated marker and endoscopic findings, and the marker characteristics. Regarding the marker characteristics, the following data were included (if available): areas under the receiver operating characteristics (ROC) curves (AUCs) and/or sensitivities and specificities with a corresponding cut-off value, and a correlation coefficient. Predictive values were not included since they are dependent on the condition prevalence, which differed between studies. For this review, the following interpretation of AUCs was adopted: AUC = 0.50–0.75 is a fair overall accuracy, AUC = 0.75–0.92 is a good overall accuracy, AUC = 0.92–0.97 is a very good overall accuracy, and AUC = 0.97–1.00 is an excellent overall accuracy. Studies exclusively reporting correlation coefficients were not included in the proper review. However, to support the associations found by others, or conversely, to contradict them, some of those studies might be discussed in the text.

Analysis of the data was conducted according to PRISMA recommendations.

## 3. Results

Our search results are presented in Figure 1. Several potentially eligible articles did not provide data allowing for the evaluation of a marker’s performance, such as AUC or sensitivity and specificity at a given cut-off value, or the marker’s performance was evaluated based on the clinical disease activity rather than the endoscopic one. As such, those articles were not included in the final synthesis. Finally, this systematic review was prepared based on 84 publications.

### 3.1. Interpretative Synthesis of Data: Fecal Markers

#### 3.1.1. Calprotectin

Although there is a wide range of biochemical, serological, immunological, genetic, epigenetic, and microbiological markers studied regarding IBD, fecal calprotectin (FC) is one of the few that was implemented in clinical practice and has been extensively used. Furthermore, it has a strong position in the international and local guidelines and can be used not only by gastroenterologists and general practitioners but also by patients at home.

Calprotectin, first described in 1980, is a 36 kDa zinc- and calcium-binding protein that belongs to the S100 family. The main source of calprotectin are neutrophils, and to a lesser extent, monocytes and macrophages. Calprotectin constitutes 60% of the soluble cytosolic proteins of neutrophils and it is used as a marker of neutrophil turnover. It can be detected in different biological fluids, such as sera, saliva, and urine, as well as in feces. The concentration of the calprotectin in stool is proportional to the neutrophils’ migration to the gastrointestinal tract, and thus, calprotectin is the most broadly measured marker in feces [27].

In clinical practice, FC measurement is employed to differentiate between functional bowel disorders, mainly irritable bowel syndrome, and inflammatory bowel disease. In patients with IBD, it is applied as a valuable non-invasive tool to monitor the disease course in terms of evaluating disease activity and mucosal healing. It was repeatedly demonstrated that FC is a sensitive marker of mucosal inflammation in IBD and is related to the extent of inflammation in UC [28,29]. In Crohn`s disease, FC (cut-off: ≥ 70 μg/g) performed more accurately in the identification of endoscopically active disease (overall accuracy: 87%) than elevated CRP, blood leukocytosis, and CDAI (overall accuracy: 66%, 54%, and 40%, respectively) [30]. Likewise, in UC, FC detected endoscopically active disease with a very good overall accuracy (89%), better than the Clinical Activity Index, elevated CRP, and blood leucocytosis (overall accuracy: 73%, 62%, and 60%, respectively) [31]. FC may additionally serve as a very good marker in the differentiation of acute severe colitis from mild to moderate colitis (cutoff value: 782 μg/g, sensitivity: 84%, specificity: 88%, AUC: 0.92) [32].

FC is a factor that predicts the treatment response [33], as well as the future course of the disease, namely relapse and postoperative recurrence. It can predict remission maintenance in CD (cutoff value: 327 µg/g, sensitivity: 92.3%, specificity: 82.4%, AUC: 0.924) [34], but also a flare in UC (cutoff value: 114 µg/g, sensitivity: 76%, specificity: 85%) [35]. In UC patients treated with infliximab, a reduction in the concentration of FC predicted the remission of disease [36].

FC has also become an integral part of the complex patient evaluation in clinical trials testing novel therapeutic agents. At the same time, home assays designed for use by patients have been developed. Smartphone-based home FC test was evaluated by 56% of patients as easy to perform and more than 90% were satisfied with it. Furthermore, there was a close correlation between laboratory and home tests [37]. Smartphone-based testing is an example of implementing an e-health strategy in real-life practice and can be a very useful alternative method of monitoring IBD patients [38].

The performance of FC in the evaluation of CD activity seems to be accurate irrespective of the disease location [39]. However, patients with ulcers limited to the ileum had significantly lower FC levels than patients with ulcerations in the colon [40]. In another study, FC correlated with the affected surface but not with the ulcerated surface and disease location [41].

#### 3.1.2. Fecal Calprotectin as a Marker of Mucosal Healing

Switching the approach to the therapeutic goals regarding IBD from clinical remission to MH resulted in the extensive assessment of FC as a non-invasive mucosal healing marker. This issue is further complicated by the definition of mucosal healing, which is far from being precise. The question arises whether an endoscopic evaluation is enough to diagnose MH, and if so, which score is optimal. Other tactics require the histologic confirmation of MH or a combination of both methods. Nevertheless, endoscopy is currently the prerequisite to diagnose mucosal healing. An alternative MH predictor is sought for this procedure, which is invasive, costly, and poorly accepted by patients.

The FC as an MH marker was evaluated in 44 eligible articles. The reported AUCs, allowing for the evaluation of the overall accuracy of a marker independently from a selected cutoff, ranged from fair to excellent overall accuracy for UC (0.70–0.99) and from fair to very good for CD (0.74–0.94) (Table 2 and Table 3, respectively). Unfortunately, a large variation in the selected optimal cutoff values was noted, ranging from 13.9 to 918 µg/g for UC and from 71 to 918 µg/g for CD. The analysis of FC’s performance as an MH marker revealed that it displayed better sensitivity regarding CD than for UC, with respective median values of 83% against 79%, and better specificity for UC than for CD, with respective median values of 85% against 71%.

Summarizing, FC may be a useful tool to detect MH; however, consensus on the cutoff value is still missing, which can be explained, at least partly, by the inter-assay variability. Carlsen et al. [42] demonstrated that FC may serve as a good indicator of deep remission at a cutoff value at ≤ 25 mg/kg, where deep remission in this study was defined using combined endoscopic (MES < 1) and histologic (Goebes Score ≤ 1) indices. Additionally, the results of the study confirmed the utility of FC as a marker of endoscopically and histologically active IBD with a cutoff of > 230 mg/kg. In turn, when the concentrations of FC are at the level of 25–230 mg/kg, researchers have suggested that endoscopy should be performed if it is needed to optimize the therapy. The performance of other markers (hemoglobin, CRP, orosomucoid, erythrocyte sedimentation rate, albumins) or the Simple Clinical Colitis Activity Index (SCCAI) as predictors of deep remission and inflammatory activity was not accurate. Jha et al. proposed 158 mg/kg for the cut-off value of FC when the definition of mucosal healing was MES ≤ 1 [43]. Moreover, in a recently published study, Walsh et al. showed that FC correlated strongly with UC activity in terms of the endoscopic (UCEIS), histologic (Nancy score), and combined evaluations. Furthermore, they identified the optimal threshold of FC concentration for histologic remission to be 72 µg/g, and for the endoscopic and combined criteria, the cutoff was 187 µg/g [46]. Zittan et al. [52] also demonstrated that low levels of FC (< 100 µg/g) closely correlated not only with endoscopic indices of IBD activity but also with histological remission (Goebes score < 3.1). Lee et al. [49] showed that the concentration of FC correlated better with UCEIS than with the Mayo score in a group of 181 UC Korean patients. Besides the correlation with endoscopic indices, a strong correlation between FC and both clinical activity index and CRP was shown. The FC evaluation had similar performance characteristics to the fecal immunochemical test (FIT); however, its specificity for identifying mucosal healing could be improved by the combined assessment of FC, FIT, and clinical symptoms [79]. Patel et al. [50] showed that there is a correlation between the FC level and disease extent but also with the worsening of activity in terms of clinical, endoscopic, and histological indices. Furthermore, these authors demonstrated that patients with mucosal healing, defined as MES ≤ 1, had significantly elevated FC if their disease was histologically active (Nancy score ≥ 2). In another study, almost 12% of patients with endoscopic remission had histologically active disease, which was accompanied by significantly elevated FC [80]. In the same study, a baseline FC level of ≥ 321 mg/g was predictive of disease relapse at the 6- and 12-month follow-up [80]. Björkesten et al. [26] analyzed a group of 64 CD patients and found that FC, in contrast to CRP, CDAI, and the Harvey–Bradshaw Index (HBI), was a good marker of MH. However, 13% of patients with endoscopically active disease had a normal FC value. Hence, they attempted to create a score by incorporating all the aforementioned indices. Despite some advantages from the combination of HBI and FC, these parameters were not superior to the FC alone. As 24% of CD patients and 15% of UC patients have undefined disease activity according to their FC level (100–250 µg/g), it has been proposed that incorporating FC into the disease activity evaluation, together with clinical indices and CRP, would serve as a better marker of disease activity than FC alone [81].

#### 3.1.3. Limitations of Fecal Calprotectin

FC may be elevated in many pathological conditions, such as gastrointestinal infections, gastric and colonic malignancies, eosinophilic colitis, lymphocytic colitis, and coeliac disease [82]. For instance, in a group of 870 outpatients referred for colonoscopy, an increased level of FC was demonstrated in 85% of patients with colorectal cancer and in 81% of patients with inflammatory conditions [83]. In turn, pregnancy does not affect the concentration of FC, and thus, its determination might be a useful non-invasive marker for monitoring the disease activity in pregnant IBD patients [84].

Multiple additional factors affect the level of FC and they all ought to be included when interpreting the result of the FC measurement.

Age, as well as concomitant medical treatment with proton pump inhibitors, nonsteroidal anti-inflammatory drugs, and acetylsalicylic acid, can all cause an increased concentration of FC. Correspondingly, Lundgren et al. [85] showed that around one-third of patients with a normal colonoscopy result had a slightly elevated FC concentration (> 50 µg/g), which was associated with the patients’ age and treatment with the above-mentioned drugs in the multivariate analysis.

Furthermore, wide daily variability of FC concentrations in stools has been observed [86]. Hence, this can be seen as a limitation for the FC application as a marker of the early response to induction therapy [87].

#### 3.1.4. Technical Considerations

Currently, several commercial assays for FC measurement are available. Studies comparing the assays demonstrate similar performance when differentiating organic and functional bowel diseases [88]. Still, there are substantial inter-assay differences between FC tests from different manufacturers, hampering the establishment of a universal cutoff value [89]. New assays for simultaneous evaluation of FC and FIT (a latex agglutination turbidimetric immunoassay system) demonstrated similar performance to a standard ELISA assay [51].

Recently, a rapid test for FC evaluation using immunochromatographic methods has been developed as a point of care test (POCT). Although semiquantitative, rapid tests yield results in 30–40 min, allowing for an immediate adjustment of the therapy if needed. As such, they offer a useful alternative to ELISA, which is the gold standard for FC evaluation but is also time-consuming and requires laboratory infrastructure. The comparison of rapid tests with ELISA demonstrated a high level of agreement between both methods [90].

#### 3.1.5. Other Fecal Markers

Among fecal markers other than calprotectin, the most intensively studied during the last decade were lactoferrin, another neutrophil-derived protein, which is responsible for limiting bacterial growth by lowering iron availability; matrix metalloproteinase (MMP)-9; and lipocalin-2. As an MH marker, lactoferrin and MMP-9 were more accurate in UC than in CD. In turn, lipocalin-2 was a more accurate marker of MH in CD (see Table 4 and Table 5). However, the number of studies is limited and the conclusion regarding the potential utility of those markers in clinical practice cannot be drawn. Moreover, the properties of the aforementioned fecal markers were not shown to be superior to FC.

### 3.2. Interpretative Synthesis of Data: Serum-Based Markers

#### 3.2.1. C-Reactive Protein (CRP)

The most intensively evaluated serum-based marker was CRP, examined in 30 eligible publications, which was frequently done as a reference for novel potential markers. CRP positively correlated with the endoscopic activity of the disease, yielding correlation coefficients from 0.29 to 0.63 for UC (Table 6) and from 0.31 to 0.71 for CD (Table 7). Reported areas under the receiver operating characteristics (ROC) curves (AUCs), allowing for the evaluation of the overall accuracy of a marker independently from the selected cut-off, ranged from fair to very good overall accuracy for both UC (0.60–0.96) and CD (0.64–0.92) (Table 6 and Table 7, respectively).

There was a very large variation in the selected optimal cut-off values, ranging from 0.4 to 28 mg/L. Differences of two orders of magnitude are unlikely for a standardized marker, such as CRP, and might imply errors in the reported units of concentration. The analysis of CRP performance as the MH marker revealed that it displayed better sensitivity than the specificity for CD, with respective median values of 79.5% against 61%, but better specificity than sensitivity for UC, with respective median values of 82% against 66%. For UC, CRP performed better when mucosal healing was defined as MES ≤ 1, or equivalent, than when restricted to MES = 0. Rosenberg et al. [106], Arai et al. [74], and Shinzaki et al. [108] compared the diagnostic power of CRP using both MH definitions. They showed CRP performance to be superior in terms of sensitivity, specificity, and/or the overall accuracy in the case of the less restrictive definition. Chen et al. [47], in turn, compared CRP as the MH marker in a cohort consisting exclusively of selected patients in clinical remission with a cohort of UC patients with clinically active or inactive disease, finding the marker’s performance to be superior in terms of specificity in the latter cohort. Accordingly, some authors have demonstrated that CRP was better at detecting moderate-to-severe disease activity but failed to differentiate between inactivity and mild endoscopic activity [30,31,78,103,105,107]. As demonstrated by Yoon et al. [110], CRP displayed similar accuracy, sensitivity, and specificity as the MH marker regardless of the type of score used for MH evaluation. However, it has to be mentioned that some authors failed to observe a significant association between CRP and endoscopic scores [32,112,113,114,115,116,117,118] and/or did not find CRP to be significantly superior to a chance marker in detecting MH in UC [42,114,119] or CD [32].

#### 3.2.2. Other Acute Phase Reactants (APRs)

CRP was not the only acute phase reactant (APR) that has been evaluated in terms of its association with the endoscopic activity of IBD. During the last decade, other positive APRs, such as procalcitonin, fibrinogen, orosomucoid, and serum amyloid A, as well as negative APRs, such as albumin and transferrin, have been appraised in this respect, although less extensively than CRP and with less success. Procalcitonin (PCT) increases in response to pro-inflammatory stimuli, particularly of bacterial origin, where the sensitivity and specificity of which as a bacterial infection marker is claimed to be superior to CRP. Accordingly, in acute UC (UCEIS ≥ 3), PCT concentrations reflected endoscopic activity more precisely than those of CRP and albumin (respectively, *r* = 0.46, *r* = 0.18, and *r* = −0.23), as reported by Wu et. al. [120], and contrary to CRP, differed significantly between MES = 2 and MES = 3, as demonstrated by Koido et al. [121]. Unfortunately, none of the authors evaluated the diagnostic power of procalcitonin as a potential MH marker. Moreover, others failed to confirm a significant association between procalcitonin and the endoscopic activity of UC [47,112,122] or CD [47,112]. Zezos et al. [123] showed that fibrinogen concentration positively correlates with endoscopic scores to a similar extent as CRP in UC (*r* = 0.49, *p* < 0.001 for both) and Eder et al. [124] in CD (respectively, *r* = 0.59 and *r* = 0.57, *p* < 0.0001 for both); however, these were not followed by the evaluation of its diagnostic power. As reported by Miranda-Garcia et al. [114], fibrinogen, with an AUC < 0.05, failed to detect MH in UC patients (Table 8) but it displayed a good overall accuracy as an MH marker, accompanied by an excellent sensitivity in one study on CD patients (Table 9). In the same study investigating fibrinogen, orosomucoid was evaluated and yielded similar overall accuracy and sensitivity but slightly lower specificity in CD, but also failed as an MH marker for UC [114]. Its unsuitability for MH detection in UC has been confirmed by Carlsen et al. [42]. Furthermore, in CD, serum amyloid A (SAA) was found to display identical good overall accuracy in two studies but superior sensitivity in one [115] and superior specificity in the other [123], probably owing to the different cutoff values applied (Table 9). For UC, SAA accuracy was found to be insufficient [42]. Albumin displayed a good overall accuracy as an MH marker for the general UC population in a study of Uchihara et al. [109] but not those of Carlsen et al. [42] or Jusué et al. [32]. For CD, albumin has been reported to inversely correlate with the SES-CD score (*r* = −0.62, *p* < 0.0001) [124] but others have found it to fail as an MH marker [32]. Moreover, even if significantly better than a chance marker, albumin seems to share similar drawbacks to CRP, i.e., being less efficient in detecting MH among patients with clinical remission [65] (Table 8). Concerning transferrin, only its correlation with MES was reported (*r* = −0.37, *p* = 0.001) [125]. Uchihara et al. [109] argued that the unsatisfactory performance of APRs in differentiating mucosal inflammation from healing results from the fact that during endoscopy, only fragments of mucosa with the most severe inflammation are considered when reporting activity scores. To support their thesis, the authors compared CRP (Table 6) and albumin (Table 8) performance as the MH markers using the traditional MES score and cumulative MES score, in which scores obtained for six different bowel regions were added. Indeed, both CRP and albumin displayed superior AUCs and specificities when the MH definition was based on the cumulative MES score.

#### 3.2.3. Cytokines, Their Receptors, and Growth Factors

CRP and other positive APRs are produced in response to cytokines, mainly interleukin (IL)-1β, IL-6, and tumor necrosis factor (TNF)-α, which also affect the concentrations of negative APRs [135]. Unsurprisingly, those cytokines, as well as many others, have also been of interest as potential MH markers, especially given that the imbalance between pro- and anti-inflammatory cytokines is a hallmark of IBD [136]. For UC, TNFα and IL-9 displayed an excellent overall accuracy, IL-6 and IL-17A displayed very good accuracy, granulocyte-macrophage colony-stimulating factor (GM-CSF) and interferon (IFN)γ displayed good accuracy, and soluble IL2 receptor (sIL2R) and IL-12(p70) displayed fair accuracy. Most of those cytokines had higher specificities than sensitivities (Table 8). However, TNFα, IL-6, IL17, and IFNγ in Rodriguez–Peralvarez et al.’s [132] study failed to detect MH and the only cytokine better than a chance marker was IL-8, displaying only a fair accuracy. IL-6 displayed an excellent overall accuracy as an MH marker for CD, IL-9 and IL-1β displayed good accuracy, and sIL2R displayed fair accuracy. The AUC for IL-17 was not calculated but the cytokine displayed good sensitivity and specificity (Table 9). While some authors reported significant correlations between the endoscopic activity of CD and IL-1, IL-6, IL-12, IFNγ, and TNFα (all *r* > 0.9, *p* < 0.0001) [137], others failed to observe significant associations for IL-6, TNFα, and IFNγ [78]. Algaba et al. [126] evaluated a panel of angiogenic factors as potential markers differentiating mucosal inflammation from healing, of which, only vascular endothelial growth factor (VEGF)-A for UC (Table 8) and angiopoietin-1 for CD (Table 9) performed significantly better than a chance marker, displaying fair overall accuracy and better sensitivities than specificities. Good and very good accuracies were reported for ST2 (interleukin 1 receptor-like 1) by two research groups for UC (Table 8) and fair accuracy by one group for CD (Table 9).

#### 3.2.4. Other Serum-Based Markers

Most of the remaining serum-based markers evaluated during the last decade as potential MH markers are also associated with inflammatory and immune responses. Several authors have evaluated lipocalin-2 (neutrophil gelatinase-associated lipocalin (NGAL)), a glycoprotein that is physiologically present in serum at low concentrations but over-secreted in response to stimuli associated with epithelial damage, either alone or in a complex with matrix metalloproteinase 9 (MMP9). NGAL displayed good overall accuracy for UC (Table 8) and fair to good accuracy for CD (Table 9). Trifoil factor (TTF)-3, a protein involved in healing of the epithelium, protecting the mucosa from insults, and the stabilization of the mucus layer, positively correlated with UCEIS [141] and displayed a fair overall accuracy as an MH marker for UC (Table 8), but failed to reflect endoscopic activity [141] or indicate MH for CD [124].

In UC, individual studies have been dedicated to adenosine deaminase activity, a non-specific marker of T-cell activation, antimicrobial peptide LL-37, adipokine and immunomodulatory cytokine visfatin (Nampt), and lysine-rich α2 glycoprotein (LRG), a protein involved in inflammatory responses to bacterial infections as well as in angiogenesis; all of these have been characterized by good overall accuracy, sensitivity, and specificity (Table 8).

While visfatin performed well for UC patients, it failed to detect MH in a mixed cohort of UC and CD patients examined by Trejo-Vasquez et al. [134]. The authors analyzed a diabetic panel including, among others, plasminogen activator inhibitor (PAI)-1 and adipokines (leptin, resistin, and visfatin), of which, only leptin displayed a fair overall accuracy as an MH marker (Table 8). Jung et al. [103] evaluated a soluble triggering receptor expressed on myeloid cells-1 (sTREM-1) in both UC and CD and found the marker to display fair sensitivity and good specificity for UC (Table 8), but with an estimated 29% sensitivity, it failed as an MH marker for CD. TREM-1 belongs to the immunoglobulin superfamily and is present on monocytes, neutrophils, macrophages, and endothelial cells, and its soluble form accompanies inflammatory and infectious conditions. Presepsin (a soluble cluster-of-differentiation 14 subtype (sCD14-ST)) is a truncated form of the co-receptor for toll-like receptor 4 (TLR4), which is expressed on phagocytic cells and shed into the circulation upon lipopolysaccharide binding in a manner proportional to the severity of the inflammatory response. Although correlated with endoscopic scores for both CD and UC (*r* = 0.277, *p* = 0.022 for UC), presepsin, in turn, differentiated mucosal inflammation and healing only for CD patients with good accuracy, sensitivity, and specificity (Table 9). Moreover, Hosomi et al. [112] demonstrated that the combined evaluation of presepsin with CRP substantially improves the sensitivity (100%) compared to individual assessments of CRP and presepsin. Similarly, regarding presepsin, the ratio of estrogen receptors (ER) β-to-α was a good MH indicator in CD (Table 9) but not UC. Solely for CD, ficolin-2, a liver-synthesized pattern recognition molecule involved in inflammation and immunity, was examined. Unlike most of the evaluated markers, its concentrations had already significantly increased in mild mucosal inflammation without a further elevation in moderate and severe inflammation, potentially making it a unique tool for differentiating mild inflammation from complete healing [77]. However, the reported overall accuracy and sensitivity have only been fair and the specificity of ficolin-2 was rather poor (Table 9). Yarur et al. [115,140] examined the performance of adhesion molecules and key proinflammatory cytokines, namely V-CAM, I-CAM, IL-1β, IL-6, IL-8, and TNFα, as potential MH markers in CD. They found that each marker’s performance in a general population of CD patients was unsatisfactory. However, as a panel, they achieved good overall accuracy (AUC = 0.81) [115]. Moreover, IL-1β and I-CAM were good predictors of MH in a subpopulation of CD patients resistant to anti-TNFα therapy [140].

### 3.3. Interpretative Synthesis of Data: Urine-Based Markers

The only eligible study evaluating potential MH markers in urine was that of Arai et al. [142], who examined prostaglandin E and found it to be a good or even excellent marker, depending on the MH definition, which showed a stepwise increase along with MES (Table 10).

## 4. Review Limitations

Although all possible efforts were made to make the review complete, some potentially relevant publications could be overlooked due to the huge number of records to be screened at the stage of the title and abstract selection. Moreover, several studies reporting marker performance were excluded upon a full-text reading due to a lack of clarity concerning whether clinical or endoscopic remission was evaluated. Furthermore, it must be pointed out that most studies included in the final analysis were conducted on small cohorts and/or had an unbalanced design with a substantial disparity between patients with endoscopic remission and inflammation, and thus the reported accuracies, sensitivities, and specificities ought to be interpreted with caution.

## 5. Conclusions

At the beginning of the 21st century, we are facing a growing incidence of IBD, especially in regions where the previously reported incidence was low, i.e., in Asia and Eastern Europe [143]. The complex nature of these diseases makes IBD patients a heterogeneous group of subjects, which can benefit from an individualized approach [144]. Still, available diagnostic and therapeutic possibilities are limited. MH has recently received interest as a therapeutic goal, for the evaluation of which, endoscopy is mandatory. At the time being, biochemical biomarkers, although possessing many advantages, cannot substitute for an invasive, expensive, and time-consuming colonoscopy. Out of those used in clinical practice, fecal calprotectin remains the closest to “the perfect” MH marker. However, as long as the standardized cutoff value of FC is missing, it cannot be broadly applied as the surrogate marker of endoscopic remission. Other fecal or serum-based markers also seem to be promising. However, as they were not evaluated in replicated and cross-validated studies and were often conducted on small cohorts, further research is required before they could substitute or be used alongside fecal calprotectin.

Ongoing research on novel biochemical biomarkers of MH should focus on validation of the most promising indices in larger and better-defined cohorts. As indicated by this review, interleukins upstream of CRP in an inflammatory cascade, namely IL-1β and IL-6, have the greatest diagnostic potential for colonic CD. For UC, IL-6, as well as TTF3, IL-9, ST2, TNFα, and ADA, seem to be more powerful than other markers. A promising tactic for getting the most out of biochemical biomarkers would be to combine multiple markers into panels, especially these displaying a high sensitivity but lower specificity with those characterized by a high specificity combined with a lower sensitivity. Such an approach already yielded excellent results for CD prediction, where a panel of 51 serum antibodies and proteins, developed on a large cohort (*n* = 400), was demonstrated as being able to predict CD diagnosis within a year with an 87% accuracy [145].

## Figures and Tables

**Figure 1 diagnostics-10-00367-f001:**
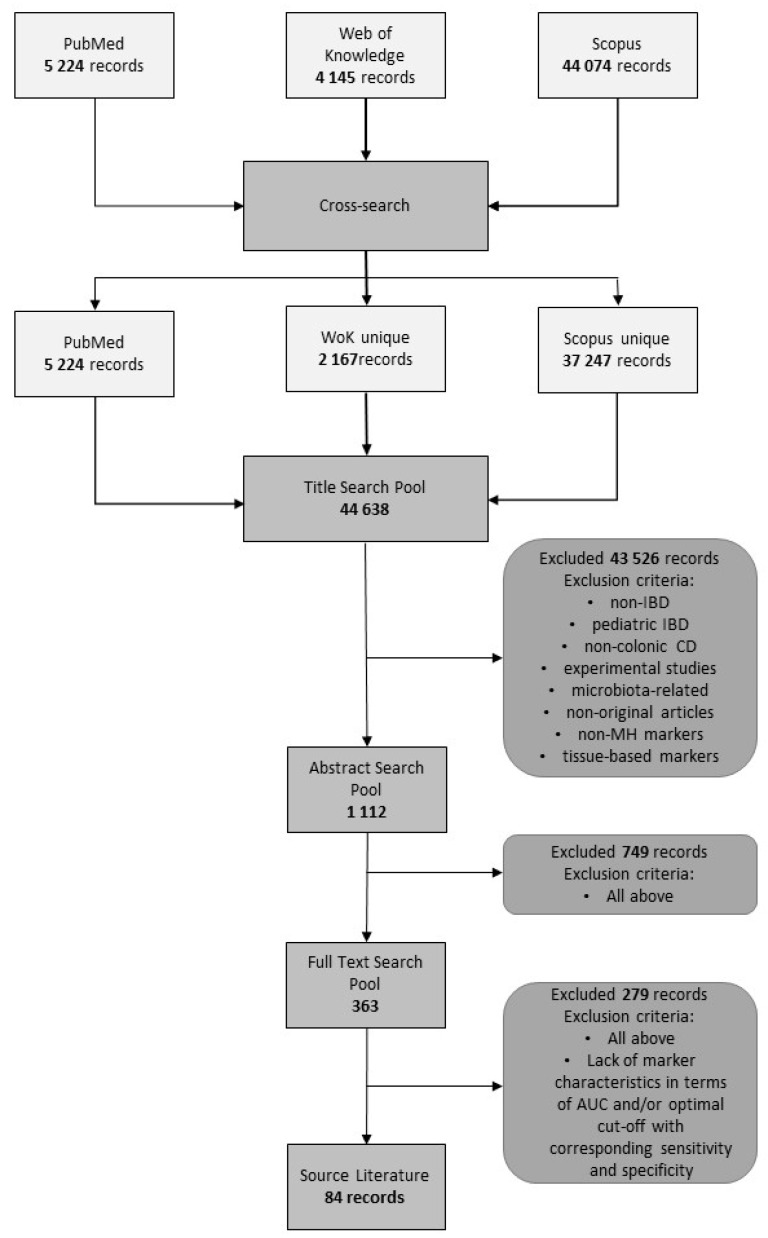
Flowchart presenting the selection process. IBD, inflammatory bowel disease; CD, Crohn’s disease; AUC, area under the receiver operating characteristics (ROC) curve.

**Table 1 diagnostics-10-00367-t001:** Selected endoscopic disease activity indices for ulcerative colitis.

	Endoscopic Activity Indices		
Parameter	MES [16]	UCEIS [21]	UCCIS [22]
Erythema	+		
Granularity			+
Vascular pattern	+	+	+
Friability	+	+	+
Erosions	+	+	+
Ulceration	+	+	+
Exudate			
MH definition	0–1	nd	nd
Validation		+	+

MES, Mayo Endoscopic Subscore; UCEIS, Ulcerative Colitis Endoscopic Index of Severity; UCCIS, Ulcerative Colitis Colonoscopic Index of Severity; MH, mucosal healing; nd, not defined.

**Table 2 diagnostics-10-00367-t002:** Diagnostic performance of fecal calprotectin for MH detection for ulcerative colitis.

Authors	*N* MH/Total	MH Definition	Marker Performance			Assay *
			AUC	Cutoff (µg/g)	Sens. and Spec.	Type, Manufacturer
Carlsen et al. [42]	68/106	MES = 0 and GS ≤ 1	0.87	≤25	58% and 90%	ELISA, Calpro Ltd.
Jha et al. [43]	5/81	MES ≤ 1	0.94	158	90% and 85%	ELiA, Phadia 100 Calprotectin
Jusué et al. [32]	30/48	MES = 0	0.9^1^ 0.9^2^	50^1^ 102^2^	79% and 85%^1^ 79% and 85%^2^	rapid kit, Quantum Blue^®^, Bühlmann ^1^low-range test; ^2^high-range test
Mak et al. [44]	23/61	MES ≤ 1	0.78	<250	77% and 67%	ELISA, Genova Diagnostics
Mine et al. [45]	45/60	MES = 0	0.77	201	71% and 78%	EliA, Immunodiahnostik AG
Walsh et al. [46]	21/66	UCEIS ≤ 1	0.92	187	100% and 67%	IBDoc^®^ FCal test, Bühlmann
Chen et al. [47]	12/44	MES ≤ 2	0.96	≤250	85% and 100%	ELISA, Bühlmann Calprotectin
Kostas et al. [48]	39/149 (mixed UC and CD)	MES = 0 ML absence (CD)	0.96	174	92% and 87%	ELISA, EK CAL, Bühlmann
Lee et al. [49]	na/181	MES = 0 UCEIS = 0	0.88	187	86% and 89%	rapid test, Quantum Blue^®^, Bühlmann
Patel et al. [50]	31/60	MES ≤ 1	0.92	60	86% and 87%	EliA, na
Hiraoka et al. [51]	75/152	MES = 0	0.8^1^ 0.82^2^	<184^1^ <224^2^	78% and 69%^1^ 79% and 78%^2^	^1^ELISA, Phical Calprotectin; ^2^LATIA
Zittan et al. [52]	44/58 (mixed UC and CD)	MES = 0 SES-CD ≤ 3	0.91	<100	71% and 91%	Buhlmann Quantum Blue Calprotectin High Range Immunoassay
Langhorst et al. [53]	72/179 colonoscopies in 91patients	RI ≤ 1	0.7	13.9	11% and 99%	*IDK*^®^ Calprotectin ELISA
Lin et al. [54]	na/52	UCEIS < 3	0.97	918	88% and 75%	ELISA, Quantum Blue Calprotectin High Range Rapid Test
Lobatón et al. [55]	35/146 colonoscopies in 123 patients	MES = 0	0.92^1^ 0.86^2^	160	67% and 85%^1^ 65% and 84%^2^	^1^Bühlmann ELISA ^2^QPOCT- Quantum Blue
Nancey et al. [56]	20/55	RI ≤ 2	0.96	250	87% and 91%	ELISA, Bühlmann
Önal et al. [57]	30/60	RI < 4	0.81	99.5 mg/L	77% and 79%	ELISA Phi Cal
Schoepfer et al. [58]	54/228	mBS ≤ 1	0.94	57	90% and 91%	ELISA Phi Cal
Falvey et al. [59]	na/65	mBS = 0	0.81	125	74% and 80%	ELISA CalPro
Kristensen et al. [60]	18/62	MES = 0	0.88	61	84% and 83%	Calpro ELISA
Kristensen et al. [60]	18/62	MES = 0	0.93	96	91% and 83%	ELISA, Bühlmann
Scaioli et al. [61]	45/121	MES = 0	0.98	110	98% and 90%	ELISA, Calprest
Takashima et al. [62]	77/105 colonoscopies in 92 patients	MES ≤ 1	0.82	200	77% and 72%	ELISA PhiCal
Theede et al. [63]	na/120	MES = 0^1^ UCEIS = 0^2^	0.88	192	75% and 88%^1^ 79% and 87%^2^	ELISA, Bülhmann
Voiosu et al. [64]	16/48 (mixed UC and CD)	MES = 0 SES-CD ≤ 3	0.77	30	94% and 50%	Buhlmann Quantum Blue Reader^®^
Yamaguchi et al. [65]	94/105 (CR)	MES = 0^1^ MES ≤ 1^2^	0.64^1^ 0.87^2^	194^1^ 200^2^	71% and 53%^1^ 67% and 91%^2^	EliA Calprotectin 2, Thermo Fisher Scientific
Nakov et al. [66]	50/116	UCEIS = 0 and MES = 0	0.988	99	97% and 98%	Quantum Blue^®^ Calprotectin, Bühlmann Laboratories AG
Hart et al. [67]	159/185	MES ≤ 1	0.722	170	65% and 69%	ELISA, Buhlmann
Karling et al. [68]	34/88	MES = 0	0.707	63	67% and 68%	CALPRO, Calprotectin ELISA Test
Yang Woon et al. [69]	7/29	MES = 0	0.88	201	82% and 100%	ELISA, Bühlmann Laboratories AG
Ryu et al. [70]	51/174	MES = 0	0.863	170	78.4% and 74.8%	ELISA, Thermo Fisher Scientific
Ryu et al. [70]	59/174	UCEIS ≤ 1	0.847	170	74.6% and 76.5%	ELISA, Thermo Fisher Scientific

*N*, number of observations; MH, mucosal healing; MES, Mayo Endoscopic Subscore; GS, Goebes Score; RI, Rachmilewitz Index; SES-CD, Simple Endoscopic Score for Crohn’s Disease; UCEIS, Ulcerative Colitis Endoscopic Index of Severity; mBS, modified Baron Score; ML, mucosal lesions; UC, ulcerative colitis; CD, Crohn’s disease; CR, subpopulation of patients in clinical remission; LATIA, latex agglutination turbidimetric immunoassay; AUC, area under receiver operating characteristics (ROC) curve; Sens., sensitivity; Spec., specificity; na, not available. *, Names of assays as provided by the authors of the cited papers. Numbers in superscript indicate parameters corresponding with a given test or MH definition.

**Table 3 diagnostics-10-00367-t003:** Diagnostic performance of fecal calprotectin in MH detection for Crohn’s disease.

Authors	*N* MH/Total	MH Definition	Marker Performance			Assay *
			AUC	Cutoff (µg/g)	Sens. and Spec.	Type, Manufacturer
Iwamoto et al. [71]	39/69	eSES-CD = 0	0.91	92	94% and 88%	EliA, Calprotectin 2
Lopes et al. [72]	19/29	SES-CD = 0	0.88	100	92% and 65%	EliA, Calprotectin
Chen et al. [47]	56/92	SES-CD	0.91	250	93% and 70%	ELISA, Bühlmann Calprotectin
Vazquez-Maron et al. [73]	22/71	SES-CD ≤ 2	na	71	96% and 52%	ELISA, Calprest^®^
Arai et al. [74]	123 colonoscopies in 89 patients	SES-CD = 0	0.81	215	83% and 71%	ELISA, PhiCal Calprotectin
Inokuchi et al. [75]	23/71	SES-CD = 0	0.82	180	87% and 71%	ELISA, Phical Calprotectin
Jusué et al. [32]	24/52	SES-CD = 0	0.7^1^ 0.7^2^	54^1^ 122^2^	63% and 71%^1^ 75% and 71%^2^	Rapid Kit, Quantum Blue^®^ Bühlmann ^1^low-range test; ^2^high-range test
Lin et al. [54]	na/36	CDEIS < 6	0.74	918	50% and 100%	ELISA, Quantum Blue Calprotectin High Range Rapid Test
Lobatón et al. [76]	40/115	CDEIS ≤ 3	0.94^1^ 0.93^2^	274^1^ 272^2^	76% and 97%	^1^Bühlmann ELISA, ^2^QPOCT-Quantum Blue
Nancey et al. [56]	40/78	SES-CD ≤ 2	0.77	250	78% and 71%	ELISA, Bühlmann
Schaffer et al. [77]	51/136	SES-CD ≤ 3	0.83	250	76% and 75%	ELISA assay RIDASCREEN^®^ CALPROTECTIN, R-Biopharm AG
Falvey et al. [59]	na/108	SES-CD ≤ 2	0.74	125	71% and 71%	ELISA CalPro
Karczewski et al. [78]	5/55	CDEIS < 3	na	76	96% and 80%	CalproLab^TM^ ELISA kit
Björkesten et al. [26]	23/126 colonoscopies in 64 patients	SES-CD ≤ 2	0.85	100	84% and 74%	ELISA, PhiCal

*N*, number of observations; MH, mucosal healing; SES-CD, Simple Endoscopic Score for Crohn’s Disease; eSES-CD, extended Simplified Endoscopic Activity Score for Crohn’s Disease (eSES-CD); CDEIS, Crohn’s Disease Endoscopic Index of Severity; AUC, area under receiver operating characteristics (ROC) curve; Sens., sensitivity; Spec., specificity; na, not available. *, Names of assays as provided by the authors of the cited papers. Numbers in superscript indicate parameters corresponding with a given test.

**Table 4 diagnostics-10-00367-t004:** Diagnostic performance of other fecal markers in MH detection for ulcerative colitis.

Marker	*N* MH/Total	MH Definition	↑/↓ in Active Disease	Correlation with Endoscopic Score	Marker Performance			Authors
					AUC	Cutoff	Sens. and Spec.	
MMP-9	na/47	MES < 1	↑	*r* = 0.653, *p* < 0.002	0.9	2.38 ng/mL	97% and 83%	Annahazi et al. [91]
MMP-9	9/32	MES ≤ 1	↑	*r* = 0.58, *p* < 0.001	0.87	900 ng/g	80% and 91%	Buisson et al. [92]
MMP-9	na/54	MES ≤ 1	↑	*r* = 0.381, *p* = 0.021	0.81	0.20 ng/mL	75% and 96%	Farkas et al. [93]
Lactoferrin	72/179 colonoscopies in 91patients	RI ≤ 1	↑	*r* = 0.4, *p* < 0.001	0.73	11.9 μg/g	75% and 63%	Langhorst et al. [53]
Lactoferrin	18/60	UCEIS = 0	↑	*r* = 0.56, *p* < 0.0001	0.71	78.3 ng/mL	57% and 88%	Mine et al. [45]
Lactoferrin	na/20	MES = 0	↑	*r* = 0.792, *p* < 0.01	0.88	288.8 ng/mL	69% and 100%	Sagawa et al. [94]
Lipocalin-2	9/32	MES ≤ 1	↑	ns	0.68	6700 ng/g	80% and 82%	Buisson et al. [92]
Lipocalin-2	265/370	MES = 0	↑	na	0.65	7 μg/g^1^ 10 μg/g^2^	40% and 77%^1^ 62% and 62%^2^	Magro et al. [95]
Lipocalin-2	UC: 15/43^1^	MES ≤ 1^1^	↑	*r* = 0.82, *p* < 0.01^1^	0.86	2.2 mg/kg	78% and 87%	Thorsvik et al. [96]
	CD: 7/30^2^	SES-CD ≤ 2^2^		*r* = 0.58, *p* < 0.01^2^				
Neopterin	20/55	RI ≤ 2	↑	*r* = 0.72, *p* < 0.001	0.98	200 pmol/g	100% and 74%	Nancey et al. [56]
F-HNP	26/45	MES ≤ 1	↑	*r* = 0.659, *p* < 0.001	0.86	32 ng/mL	96% and 74%	Kanmura et al. [97]
PMNE	72/179 colonoscopies in 91 patients	RI ≤ 1	↑	*r* = 0.38, *p* < 0.001	0.7	0.035 μg/mL	39% and 87%	Langhorst et al. [53]

*N*, number of observations; MH, mucosal healing; **↑/↓,** increased/decreased; *r*, correlation coefficient; *p*, *p*-value; UC, ulcerative colitis; CD, Crohn's disease; MES, Mayo Endoscopic Subscore; UCEIS, Ulcerative Colitis Endoscopic Index of Severity; RI, Rachmilewitz Index; SES-CD, Simple Endoscopic Score for Crohn’s Disease; MMP-9, matrix metalloproteinase 9; F-HNP, fecal human neutrophil peptides; PMNE, polymorphonuclear neutrophil elastase; FIT, fecal immunochemical test; AUC, area under receiver operating characteristics (ROC) curve; Sens., sensitivity; Spec., specificity; na, not available; ns, not significant. Numbers in superscript indicate parameters corresponding with a given cohort or selected cutoff.

**Table 5 diagnostics-10-00367-t005:** Diagnostic performance of other fecal markers in MH detection for Crohn’s disease.

Marker	*N* MH/Total	MH Definition	↑/↓ in Active Disease	Correlation with Endoscopic Score	Marker Performance			Authors
					AUC	Cutoff	Sens. and Spec.	
MMP-9	31/54	CDEIS no ulceration	↑	*r* = 0.55, *p* < 0.001	0.72	350 ng/g	64% and 90%	Buisson et al. [92]
MMP-9	na/50	SES-CD ≤ 4	↑	na	ns			Farkas et al. [93]
Lactoferrin	na/101	SES-CD ≤ 3	↑	*r* = 0.5, *p* < 0.0001	0.68	145.82 μg/mL	85% and 61%	Klimczak et al. [98]
Lipocalin-2	31/54	CDEIS no ulceration	↑	*r* = 0.49, *p* < 0.001	0.68	67000 ng/g	46% and 86%	Buisson et al. [92]
Neopterin	40/78	SES-CD ≤ 2	↑	*r* = 0.47, *p* < 0.001	0.75	200 pmol/g	73% and 74%	Nancy et al. [56]

*N*, number of observations; MH, mucosal healing; **↑/↓,** increased/decreased; *r*, correlation coefficient; *p*, *p*-value; SES-CD, Simple Endoscopic Score for Crohn’s Disease; CDEIS, Crohn’s Disease Endoscopic Index of Severity; MMP-9, matrix metalloproteinase 9; AUC, area under receiver operating characteristics (ROC) curve; Sens., sensitivity; Spec., specificity; na, not available; ns, not significant.

**Table 6 diagnostics-10-00367-t006:** Diagnostic performance of C-reactive protein in MH detection for ulcerative colitis.

N MH/Total	MH Definition	Correlation with Endoscopic Score	Marker Performance			Authors
			AUC	Cutoff	Sens. and Spec.	
47/79^1^ 66/79^2^	MES = 0^1^ MES ≤ 1^2^	*r* = 0.61, *p* < 0.05	0.77^1^ 0.96^2^	0.05 mg/dL^1^ 0.39 mg/dL^2^	71% and 71%^1^ 92% and 92%^2^	Arai et al. [74]
20/43	EAI ≤ 3	na	0.80	3.8 mg/L	90% and 67%	Beyazit et al. [99]
6/34	MES = 0	*r* = 0.35, *p* = 0.07	na	5 mg/L	65% and 50%	Bodelier et al. [81]
12/19 (CR)^1^ 12/45 (all)^2^	MES ≤ 2	*r* = 0.634, *p* < 0.001	0.86^1^ 0.81^2^	5 mg/L	100% and 23%^1^ 100% and 42%^2^	Chen et al. [47]
28/66 (NT)^1^ 88/132 (T)^2^ pooled^3^	MES ≤ 1	*r* = 0.317, *p* < 0.01^1^ *r* = 0.408, *p* < 0.01^2^ *r* = 0.372, *p* < 0.01^3^	0.701 0.802 0.773	5 mg/L^1^ 7.2 mg/L^2^ 7.3 mg/L^3^	na and 82%^1^ 44% and 93%^2^ 39% and 91%^3^	de Bruyn et al. [100]
48/103	MES ≤ 1	*r* = 0.326, *p* = 0.001	0.60	0.7 mg/L	52% and 69%	Dranga et al. [101]
29/44	MES ≤ 1	na	0.77	28 mg/L	71% and 85%	Hassan et al. [102]
10/85	MES = 0	*r* = 0.386, *p* < 0.001	na	0.5 mg/dL	33% and 100%	Jung et al. [103]
30/48	MES = 0	ns	0.7	0.25 mg/dL	81% and 53%	Jusué et al. [32]
72/179 colonoscopies in 91 patients	RI ≤ 1	*r* = 0.29, *p* < 0.001	0.65	0.25 mg/dL	46% and 82%	Langhorst et al. [53]
16/53	MES = 0	*r* = 0.46, *p* < 0.001	0.673	0.5 mg/L	54% and 83%	Neubauer et al. [104]
82/149	MES = 0^1^ MES ≤ 1^2^	na	0.62^1^ 0.76^2^	1.6 mg/L^1^ 2.2 mg/L^2^	58% and 66%^1^ 65% and 82%^2^	Rosenberg et al. [105]
7/32	MES ≤ 1	*r* = 0.387, *p* < 0.05	0.77	na	na	Samant et al. [106]
34/134	RI ≤ 3	*r* = 0.503, *p* < 0.001	na	5 mg/L	67% and 60%	Schoepfer et al. [30]
54/228	mBS ≤ 1	*r* = 0.556, *p* < 0.001	0.78	6 mg/L	72% and 68%	Schoepfer et al. [58]
16/129^1^ 50/129^2^	Matts’ = 1^1^ Matts’ ≤ 2^2^	na	0.67^1^ 0.81^2^	na	na	Shinzaki et al. [107]
9/34 (na)*	MES = 0	na	0.71	0.4 mg/L	71% and 30%	Tran et al. [108]
52/207	mMES ≤ 1^1^ cMES ≤ 8^2^	*r* = 0.38, *p* < 0.05^1^ *r* = 0.52, *p* < 0.05^2^	0.75^1^ 0.90^2^	0.23 mg/dL	81% and 57%^1^ 81% and 85%^2^	Uchihara et al. [109]
94/105 (CR)	MES = 0^1^ MES ≤ 1^2^	na	0.62^1^ 0.74^2^	0.08 mg/dL^1^ 0.04 mg/dL^2^	na 68% and 70%^2^	Yamaguchi et al. [65]
na/65	mBS = 0	*r* = 0.45, *p* < 0.001	0.68	5 mg/L	80% and 50%	Falvey et al. [59]
722 endoscopies in 552 patients	P-T = 0^1^ MES ≤ 1^2^ mBS = 0^3^ HSI ≤ 4^4^ RI ≤ 4^5^	*r* = 0.457, *p* < 0.001^1^ *r* = 0.503, *p* < 0.001^2^ *r* = 0.520, *p* < 0.001^3^ *r* = 0.507, *p* < 0.001^4^ *r* = 0.523, *p* < 0.001^5^	0.76^1^ 0.77^2^ 0.78^3^ 0.77^4^ 0.76^5^	8 mg/L	51% and 85%^1^ 53% and 84%^2^ 52% and 87%^3^ 51% and 85%^4^ 51% and 87%^5^	Yoon et al. [110]

*N*, number of observations; MH, mucosal healing; *r*, correlation coefficient; *p*, *p*-value; CR, subgroup of patients in clinical remission; NT, not treated with infliximab; T, treated with infliximab; MES, Mayo Endoscopic Subscore; UCEIS, Ulcerative Colitis Endoscopic Index of Severity; RI, Rachmilewitz Index; mBS, modified Baron Score; EAI, Endoscopic Activity Index; P-T, Powell-Tuck Assessment; HSI, Hanauer’s Sigmoidoscopic Index; AUC, area under receiver operating characteristics (ROC) curve; Sens., sensitivity; Spec., specificity; na, not available; ns, not significant; *, counted from a scatter-plot. Numbers in superscript indicate parameters corresponding with a given MH definition or cohort.

**Table 7 diagnostics-10-00367-t007:** Diagnostic performance of C-reactive protein in MH detection for Crohn’s disease.

N MH/Total	MH Definition	Correlation with Endoscopic Score	Marker Performance			Authors
			AUC	Cutoff	Sens. and Spec.	
42/209 endoscopies	SES-CD ≤ 2	*r* = 0.56, *p* < 0.001	0.64	3 mg/L	50% and 24%	Björkesten et al. [26]
31/50	SES-CD ≤ 3	*r* = 0.45, *p* = 0.07	na	5 mg/L	65% and 56%	Bodelier et al. [81]
20/34 (CR)^1^ 21/56 (all)^2^	SES-CD ≤ 3	*r* = 0.658, *p* < 0.01	0.76^1^ 0.81^2^	5 mg/L	83% and 46%^1^ 91% and 71%^2^	Chen et al. [47]
38/108	Descriptive corresponding with SES-CD ≤ 2	*r*= 0.307, *p* < 0.001	0.74	5 mg/L	79% and 57%	de Bruyn et al. [111]
na/107	SES-CD ≤ 2	*r* = 0.44, *p* < 0.001	0.64	5 mg/L	67% and 60%	Falvey et al. [59]
na/33	SES-CD = 0	*r* = 0.709, *p* < 0.001	0.92	0.03 mg/dL	86% and 89%	Hosomi et al. [112]
19/55	SES-CD ≤ 3	*r* = 0.61, *p* < 0.01	0.75	0.11 mg/dL	68% and 78%	Ishihara et al. [113]
6/34	SES-CD ≤ 3	*r* = 0.585, *p* < 0.001	na	0.4 mg/dL	89% and 67%	Jung et al. [104]
4/55	CDEIS ≤ 2	*r* = 0.672, *p* < 0.001	na	3 mg/L	80% and 76%	Karczewski et al. [78]
11/43	Descriptive corresponding with SES-CD ≤ 3	na	0.78	1.1 mg/L	100% and 38%	Miranda-García et al. [114]
51/136	SES-CD ≤ 3	*r* = 0.458, *p* < 0.001	0.69	5 mg/L	84% and 53%	Schaffer et al. [77]
26/140	SES-CD ≤ 3	*r* = 0.53, *p* < 0.01	na	5 mg/L	58% and 68%	Schoepfer et al. [30]
39/94	Descriptive corresponding with SES-CD ≤ 2	na	0.75	4 mg/L	69% and 62%	Yarur et al. [115]

*N*, number of observations; MH, mucosal healing; *r*, correlation coefficient; *p*, *p*-value; CR, subgroup of patients in clinical remission; SES-CD, Simple Endoscopic Score for Crohn’s Disease; CDEIS, Crohn’s Disease Endoscopic Index of Severity; MH, mucosal healing; AUC, area under receiver operating characteristics (ROC) curve; Sens., sensitivity; Spec., specificity; na, not applicable. Numbers in superscript indicate parameters corresponding with a given cohort.

**Table 8 diagnostics-10-00367-t008:** Diagnostic performance of other serum-based markers in MH detection for ulcerative colitis.

Marker	*N* MH/Total	MH Definition	↑/↓ in Active Disease	Correlation with Endoscopic Score	Marker Performance			Authors
					AUC	Cutoff	Sens. and Spec.	
VEGF-A	16/37	MES = 0	↑	*r* = 0.397, *p* = 0.015	0.72	341 ng/mL	64% and 85%	Algaba et al. [126]
ADA	20/43	EAI ≤ 3	↑	na	0.87	9.45 U/L	84% and 83%	Beyazit et al. [99]
ST2	25/83	EAI ≤ 4	↑	na	0.81	47.1 pg/mL	72% and 72%	Boga et al. [127]
ST2	44/84	MES ≤ 1	↑	*r* = 0.762, *p* < 0.001	0.92	74.87 ng/L	83% and 83%	Díaz-Jiménez et al. [128]
ST2	18/24	MES ≤ 1	↑	*r* = 0.66, *p* < 0.001	na	74.87 ng/L	44% and 95%	Díaz-Jiménez et al. [129]
NGAL	14/41	MES ≤ 1	↑	*r* = 0.574, *p* < 0.05	0.76	43.6 ng/mL	96% and 54%	Budzynska et al. [130]
NGAL-MMP9 complex	28/66 (NT)^1^ 88/132 (T)^2^ pooled^3^	MES ≤ 1	↑	*r* = 0.317, *p* < 0.01^1^ *r* = 0.382, *p* < 0.01^2^ *r* = 0.37, *p* < 0.01^3^	0.75^1^ 0.78^2^ 0.78^3^	97.7 ng/mL^1^ 93.2 ng/mL^2^ 97.7 ng/mL^3^	43% and 93%^1^ 44% and 91%^2^ 43% and 91%^3^	de Bruyn et al. [100]
sIL2R	na/68	MES = 0	↑	*r* = 0.357, *p* = 0.003	0.65	274 U/mL	60% and 76%	Hosomi et al. [112]
sTREM-1	10/85	MES = 0	↑	*r* = 0.498, *p* < 0.001	na	60 pg/mL	59% and 80%	Jung et al. [103]
IL-6	5/45	MES = 0	↑	*r* = 0.596, *p* < 0.001	0.93	9.6 pg/mL	80% and 95%	Mankowska-Wierzbicka et al. [116]
IL-17	5/45	MES = 0	↑	*r* = 0.578, *p* < 0.001	0.93	6.6 pg/mL	60% and 92%	Mankowska-Wierzbicka et al. [116]
TNFα	5/45	MES = 0	↑	*r* = 0.701, *p* < 0.001	0.98	7.6 pg/mL	80% and 95%	Mankowska-Wierzbicka et al. [116]
IL-9	na/53	MES = 0	↑	*r* = 0.74, *p* < 0.001	0.97	20.5 pg/mL	94% and 92%	Matusiewicz et al. [119]
GM-CSF	16/53	MES = 0	↑	*r* = 0.56, *p* < 0.001	0.91	16.7 pg/mL	69% and 97%	Neubauer et al. [104]
IFNγ	16/53	MES = 0	↑	*r* = 0.55, *p* < 0.001	0.78	83.2 pg/mL	100% and 60%	Neubauer et al. [104]
IL-12(p70)	16/53	MES = 0	↑	*r* = 0.49, *p* < 0.001	0.71	21.6 pg/mL	50% and 100%	Neubauer et al. [104]
Nampt	66/98	MES ≤ 1	↑	*r* = 0.47, *p* < 0.001	0.77	1.54 ng/mL	76% and 75%	Neubauer et al. [131]
IL-8	22/67	BS ≤ 1 (inactive + mild)	↑	na	na	13.74 pg/mL	69% and 55%	Rodriguez-Peralvarez et al. [132]
LRG	16/129^1^ 50/129^2^	Matts’ = 1^1^ Matts’ ≤ 2^2^	↑	na	0.76^1^ 0.85^2^	na	na	Shinzaki et al. [107]
TTF3	43/76 (CR)	BS ≤ 1	↑	na	0.73	1.27 ng/mL	70% and 68%	Srivastava et al. [133]
TTF3	50/116	UCEIS = 0 and MES = 0	↑	*r* = 0.82 for UCEIS, *r* = 0.811 for EMS, *p* < 0.001	0.927	6.74 ng/mL	89.9% and 86.9%	Nakov et al. [66]
LL-37	9/34 (na)*	MES = 0	↓	na	0.76	32 mg/mL	100% and 38%	Tran et al. [108]
Leptin	8/23 UC^1^ 5/11 CD^2^	MES = 0^1^ SES-CD = 0^2^	↓	na	0.65	2.5 ng/mL	88% and 45%	Trejo-Vasquez et al. [134]
ALB	52/207	mMES ≤ 1^1^ cMES ≤ 8^2^	↓	*r* = −0.52, *p* < 0.05^1^ *r* = −0.65, *p* < 0.05^2^	0.77^1^ 0.90^2^	4.2 g/dL	73% and 72%^1^ 79% and 88%^2^	Uchihara et al. [109]
ALB	94/105 (CR)	MES = 0^1^ MES ≤ 1^2^	↓	na	0.50^1^ 0.55^2^	4.3 g/dL	na 34% and 90%	Yamaguchi et al. [65]

*N*, number of observations; MH, mucosal healing; **↑/↓,** increased/decreased; *r*, correlation coefficient; *p*, *p*-value; CR, subgroup of patients in clinical remission; NT, not treated with infliximab; T, treated with infliximab; (m)MES, Mayo Endoscopic Subscore; c(MES), cumulative Mayo Endoscopic Score; UCEIS, Ulcerative Colitis Endoscopic Index of Severity; RI, Rachmilewitz Index; EAI, Endoscopic Activity Index; SES-CD, Simple Endoscopic Score for Crohn’s Disease; VEGF-A, vascular endothelial growth factor-A; ADA, adenosine deaminase; ST2, decoy receptor for IL-33 (member of IL-1 receptor family); NGAL, neutrophil gelatinase B–associated lipocalin; NGAL-MMP9, neutrophil gelatinase B–associated lipocalin-matrix metalloproteinase 9; sIL2R, soluble IL-2 receptor; sTREM-1, soluble triggering receptor expressed on myeloid cells-1; IL, interleukin; TNFα, tumor necrosis factor α; GM-CSF, granulocyte-macrophage colony-stimulating factor; IFNγ, interferon-γ; Nampt, nicotinamide phosphoribosyltransferase; LRG, leucine-rich alpha-2 glycoprotein; TTF3, trefoil factor-3; LL-37, cathelicidin; ALB, albumins; AUC, area under receiver operating characteristics (ROC) curve; Sens., sensitivity; Spec., specificity; na, not available; *, counted from a scatterplot. Numbers in superscript indicate parameters corresponding with a given MH definition or cohort.

**Table 9 diagnostics-10-00367-t009:** Diagnostic performance of other serum-based markers in MH detection in Crohn’s disease.

Marker	*N* MH/Total	MH Definition	↑/↓ in Active Disease	Correlation with Endoscopic Score	Marker Performance			Authors
					AUC	Cutoff	Sens. and Spec.	
Ang1	19/35	SES-CD ≤ 2	↑	*r* = 0.362, *p* = 0.049	0.65	47.8 ng/mL	52% and 67%	Algaba et al. [126]
ST2	52/8	SES-CD ≤ 2	↑	na	0.73	57.6 pg/mL	100% and 56%	Boga et al. [127]
NGAL	45/79	SES-CD ≤ 7 (inactive + mild)	↑	ns	0.61	72.5 ng/mL	48% and 83%	Budzynska et al. [130]
NGAL-MMP9 complex	38/108	Descriptive corresponding with SES-CD ≤ 2	↑	*r*= 0.296, *p* < 0.001	0.77	45 ng/mL	82% and 60%	de Bruyn et al. [111]
IL-9	23/50	Descriptive corresponding with CDEIS ≤ 2	↑	*r* = 0.395, *p* < 0.001	0.78	18.1 pg/mL	52% and 100%	Feng et al. [138]
sIL2R	na/33	SES-CD = 0	↑	*r* = 0.516, *p* = 0.002	0.74	283 U/mL	71% and 73%	Hosomi et al. [112]
sCD14-ST	na/33	SES-CD = 0	↑	*r* = 0.512, *p* = 0.002	0.85	89 pg/mL	86% and 85%	Hosomi et al. [112]
SAA	19/55	SES-CD ≤ 3	↑	*r* = 0.64, *p* < 0.01	0.77	5.9 μg/dL (mL)*	68% and 83%	Ishihara et al. [113]
SAA	39/94	Descriptive corresponding with SES-CD ≤ 2	↑	na	0.77	28 mg/L	95% and 64%	Yarur et al. [115]
sTREM-1	6/34	SES-CD ≤ 3	↑	ns	Na	60 pg/mL	29% and 83%	Jung et al. [103]
IL-17	4/55	CDEIS ≤ 2	↑	*r* = 0.296, *p* < 0.005	Na	7.05 pg/mL	80% and 74%	Karczewski et al. [78]
ERβ/ERα	11/31	SES-CD ≤ 2		na	0.84	0.85 (ratio)	91% and 70%	Linares et al. [139]
IL-6	6/32	CDEIS ≤ 7	↑	*r* = 0.957, *p* < 0.001	1.00	15.9 pg/mL	100% and 100%	Ljuca et al. [137]
Orosomucoid	11/43	Descriptive corresponding with SES-CD ≤ 3	↑	na	0.85	119.5 mg/dL	100% and 57%	Miranda-García et al. [114]
Fibrinogen	11/43	Descriptive corresponding with SES-CD ≤ 3	↑	na	0.81	457 mg/dL	100% and 65%	Miranda-García et al. [114]
Ficolin-2	51/136	SES-CD ≤ 3	↑	*r* = 0.171, *p* = 0.047	0.61	2.404 μg/mL	51% and 68%	Schaffer et al. [87]
IL-1β	25/47 (RP)	Descriptive corresponding with SES-CD ≤ 2	↑	na	0.88	na	na	Yarur et al. [140]
ICAM	25/47 (RP)	Descriptive corresponding with SES-CD ≤ 2	↑	na	0.89	na	na	Yarur et al. [140]

*N*, number of observations; MH, mucosal healing; **↑/↓,** increased/decreased; *r*, correlation coefficient; *p*, *p*-value; RP, patients resistant to anti-TNF treatment; SES-CD, Simple Endoscopic Score for Crohn’s Disease; CDEIS, Crohn’s Disease Endoscopic Index of Severity; Ang1, angiopoietin 1; ST2, decoy receptor for IL-33 (a member of the IL-1 receptor family); NGAL, neutrophil gelatinase B–associated lipocalin; NGAL-MMP9, neutrophil gelatinase B–associated lipocalin-matrix metalloproteinase 9; IL, interleukin; sIL2R, soluble IL-2 receptor; sCD14-ST, soluble CD14 subtype (presepsin); SAA, serum amyloid A; sTREM-1, soluble triggering receptor expressed on myeloid cells-1; ERβ/ERα, ratio of circulating estrogen receptors beta and alpha; ICAM, intercellular adhesion molecule 1; AUC, area under receiver operating characteristics (ROC) curve; Sens., sensitivity; Spec., specificity; na, not available; ns, not significant; * different units stated throughout the article.

**Table 10 diagnostics-10-00367-t010:** Diagnostic performance of urine-based markers in MH detection for ulcerative colitis.

Marker	*N* MH/Total	MH Definition	↑/↓ in Active Disease	Correlation with Endoscopic Score	Marker Performance			Authors
					AUC	Cutoff	Sens. and Spec.	
PGE	47/79^1^ 66/79^2^	MES = 0^1^ MES ≤ 1^2^	↑	*r* = 0.85, *p* < 0.05	0.90^1^ 0.98^2^	21.8 μg/g CR^1^ 34.7 μg/g CR^2^	81% and 81%^1^ 89% and 89%^2^	Arai et al. [142]

*N*, number of observations; MH, mucosal healing; **↑/↓,** increased/decreased; *r*, correlation coefficient; *p*, *p*-value; PGE, prostaglandin E; CR, creatinine; MES, Mayo Endoscopic Score; AUC, area under receiver operating characteristics (ROC) curve; Sens., sensitivity; Spec., specificity. Numbers in superscript indicate parameters corresponding with a given MH definition.

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
