# Peer review of "Biochemical Biomarkers of Mucosal Healing for Inflammatory Bowel Disease in Adults"

_diagnostics, 2020, doi:10.3390/diagnostics10060367_

Round 1

Reviewer 1 Report

Very informative and well conducted review .84 publications were considered. The  clinical and biological parameters  chosen  by the authors in the context  of mucosal healing are well argued . Therefore  , the autors omitted the  cytokine system  as a potentiel  biomarkers  and potential monitoring of IBD patients . In this sense, the authors must add one ore two sentences ( Soufli et al, 2016, Rafa et al, 2013; 2017).

What  about using Probiotics or their metabolites in therapies?

Reviewer 2 Report

This is a concise review on the biomarkers of mucosal healing in inflammatory bowel disease. However, I do not understand why this is a systematic review, as the authors have not performed a meta-analysis of the various studies, but rather a review of what has been published so far.

Comments

1. It is not clear how the authors came down from 53443 publications to reviewing only 84. A number of markers have been missed, such as erythrocyte sedimentation rate, serum alpha-1 antitrypsin, serum hepcidin, ferritin, anti-glycoprotein 2, anti-granulocyte macrophage colony-stimulating factor, anti-neutrophil cytoplasmic antibody, anti-Saccharomyces cerevisiae antibody, anti-laminaribioside carbohydrate IgG antibody, anti-mannobioside carbohydrate IgG antibody, antibody to the outer membrane protein of Escherichia coli, anti-CBir1, anti-chitobioside (ACCA), anti-chitin (anti-C) antibodies, anti-outer membrane porin C (anti-OmpC), anti-12 antibody. Also, autoantibodies targeting the exocrine pancreas (PAB) were shown to be highly specific for CD. In contrast, UC has been associated with anti-neutrophil cytoplasmic autoantibodies (pANCA) and antibodies against goblet cells (GAB). Although I understand that the review focuses on mucosal healing and data on all the above-mentioned markers may not be sufficient, the authors should add a section on the ongoing research in finding all those new surrogate markers that could help us replace endoscopy while maintaining an accurate diagnosis.

2. The manuscript contains numerous typos which should be corrected by a native English speaker.

3. The authors should pay attention to abbreviations. They should explain it the first time they appear in the text and they should use only the abbreviated form thereafter.

Reviewer 3 Report

In this manuscript, the authors wrote an excellent and very comprehensive systematic review on the performance of fecal-, serum- and urine-based biochemical markers in “reflecting” mucosal healing (defined by endoscopy). Its outstanding importance relies in trying to find out potential non-invasive markers, which could replace endoscopy. Especially during this pandemic period, these markers could be very useful. Very good choice of the topic! Although the authors performed a very thorough search of the literature (main databases), the standard known markers, like fecal calprotectin and serum C-reactive protein were the most used markers. However, even though it was the best marker, not even fecal calprotectin was found to be able to replace colonoscopy. I appreciate the immense efforts of the authors, the accuracy of the data, all the details presented in the tables and the in depth analysis of results. The manuscript is a bit long, however easily to be read.

Other comments/suggestions:

  1. Introduction a. Line 22 – please insert “potentially” before devastating. b. Line 33 – please correct “heterogonous”. c. Line 35 – please insert also “epigenetical”, among factors. d. Lines 42-44 – Please inset the following references for MH in CD: 1. Peyrin-Biroulet L, Sandborn W, Sands BE, et al. Selecting therapeutic targets in inflammatory bowel disease (STRIDE): Determining therapeutic goals for treat-to-target. Am J Gastroenterol 2015; 110: 1324-1338. 2. Shah SC, Colombel JF, Sands BE, Narula N. Systematic review with meta-analysis: mucosal healing is associated with improved long-term outcomes in Crohn’s disease. Aliment Pharmacol Ther 2016; 43: 317-333. 3 Reinink AR, Lee TC, Higgins PD. Endoscopic mucosal healing predicts favorable clinical outcomes in inflammatory bowel disease: A meta-analysis. Inflamm Bowel Dis 2016; 22: 1859-1869. e. Lines 61-63 – Instead of reference 3, please use the following: 1. Maaser C, Sturm A, Vavricka SR, et al. ECCO-ESGAR guideline for diagnostic assessment in IBD part 1: initial diagnosis, monitoring of known IBD, detection of complications. J Crohns Colitis 2019; 13: 144–164. 2. Sturm A, Maaser C, Calabrese E, et al. ECCO-ESGAR Guideline for Diagnostic Assessment in IBD Part 2: IBD scores and general principles and technical aspects. J Crohns Colitis 2019; 13: 273–284. f. Also, reference 16 is wrongly used – it does not refer to the Japanese Guidelines; that would be reference 17. Please correct. g. Line 71 –reference 19 is about UCEIS, not UCCIS. For UCCIS please use the following reference: Samuel S, Bruining DH, Loftus EV Jr, et al. Validation of the ulcerative colitis colonoscopic index of severity and its correlation with disease activity measures. Clin Gastroenterol Hepatol 2013;11:49–54.e1. h. Table 1 - Endoscopic disease activity indices for ulcerative colitis. I suggest here to be presented only MES, UCEIS and UCCIS. There is no point to show the other indices. MES is the most used, while UCEIS and UCCIS are validated. They should be used. This manuscript should not present the history of all endoscopy indices. They are already mentioned in various guidelines, belonging to different societies across the world. i. Lines 84-86: please insert here also ultrasonography – simple or with contrast – SICUS and CEUS. These techniques are also included in the guidelines. CT scan is less indicated nowadays (and it should be avoided as much as possible), due to the high level of irradiation, increasing the cancer risk in IBD patients, who have already implicitly a high risk of cancer. j. It appears that the references should be reviewed carefully, as there are many errors. Line 89 – reference 24 is wrong. It should be reference 25. Line 90 – for SES-CD, the correct reference is Daperno M, D’Haens G, Van Assche G, et al. Development and validation of a new, simplified endoscopic activity score for Crohn’s disease: the SES-CD. Gastrointest Endosc 2004;60:505–12. This references does not appear at all. k. I do not see the point in detailing all endoscopic indices (type of lesions included etc), since it is not the aim of this review. They should only be mentioned and pointing out that there is no universal consensus regarding which one to use and the definition of MH. l. Since the authors mentioned ileocolonoscopy, a sentence should be dedicated to the upper endoscopy in CD, which is also necessary to assess the lesions proximal to terminal ileum…and cannot be assessed by imaging.
  2. Methods. a. I suggest as terms for queries also “ileal inflammation”, “small bowel inflammation” and “endoscopy”. b. Why did the authors exclude “pediatric IBD”? IBD in children represents a very important issue. Non-invasive markers are required much more than in adults and there is a plethora of excellent articles. It should be mentioned from the beginning (title and abstract) that the included population is represented by adults only. c. Why did the authors exclude “non-colonic CD”? Isolated CD (L1) could have been very useful to be included as well and to study the correlation between MH and the presented markers. However, since this is what the authors decided, they should mention from the beginning (including the abstract) that they took into consideration only manuscripts regarding colonic IBD (UC and colonic CD). Also, from what I understand, IBD-U was not included. d. Line 139: what do the authors understand by mixed cohort? e. Line 141 – scale or score used? Please revise.
  3. Results: a. Figure 1 is mentioned both in Methods and Results. Results should then refer to Figure 1, without repeating data from the Figure 1 (in the text). b. Lines 164-166 – please insert also “epigenetical” (miRNAs are very important - and they are not the only epigenetical factors studied) i.e. Berezin, A., & Poplyonkin, E. (2020). Diagnostic and therapeutic value of micro-RNAs in inflammatory bowel disease. Biomedical Research and Therapy, 7(2), 3622-3632. c. Line 212: please add “s”: “over 70% of gastroenterologist” d. Lines 212-213- not relevant; please delete; we are in 2020, not 2011. It is already clear that clinical activity is not enough. e. Please correct ref. 41 – the name is Jusué (also in Table 2 and Table 3 and Table 6). f. Please correct name in Table 2 – Walsh, not Welsh. g. Please correct name in ref. 71 - Scaioli (not Scaiolia). g. I appreciate the “Limitations of FC”, as well as “Technical considerations”– well described. h. It should be a subtitle 3.2. Interpretative synthesis of data: serum-based markers, before 3.2.1 C-reactive protein (since there are subtitles 3.1 and 3.3). i. 3.2.4. Other serum-based markers: it should be “3.2.3”. j. Tables 8 and 9 are of particular interest, for future studies. Very good synthesis in the text of the most important data shown in Tables.
  4. Conclusion. a. Preferably, it should not include references, as it should contain the concluded remarks of the whole paper. In any case, instead of the present ref. 152, a more recent one should be used: Ng SC, Shi HY, Hamidi N, et al. Worldwide incidence and prevalence of inflammatory bowel disease in the 21st century: a systematic review of population-based studies. Lancet. 2018;390(10114):2769‐2778. b. Conclusion should also emphasize more the possible role of the other serum-based markers (and which ones), as directions for future research, since some of them showed good accuracy as markers for mucosal healing in UC and/or CD.

Round 2

Reviewer 2 Report

Please take care of your references inside your manuscript. For example, do not write 9,10,11, but 9-11. Also, do not write 108,104,78 but you should write 78,104,108, and so on. They should all be in ascending order.
